# A Roadmap for Gigabit to Terabit Optical Wireless Communications Receivers

**DOI:** 10.3390/s23031101

**Published:** 2023-01-18

**Authors:** William Matthews, Steve Collins

**Affiliations:** Department of Engineering Science, University of Oxford, Oxford OX1 3PJ, UK

**Keywords:** visible light communications, optical wireless communications, SiPM, Monte Carlo simulation

## Abstract

Silicon photomultipliers’ relatively large areas and ability to detect single photons make them attractive as receivers for optical wireless communications. In this paper, the relative importance of the non-linearity and width of SiPMs’ fast output in their performance in receivers is investigated using Monte Carlo simulations. Using these results, the performances of receivers containing different SiPMs are estimated. This is followed by a discussion of the potential performances of arrays of existing SiPMs. Finally, the possible dramatic improvements in performance that could be achieved by using two stacked integrated circuits are highlighted.

## 1. Introduction

Optical wireless communications (OWCs) and visible light communications (VLCs) systems are being investigated as ways to increase local wireless communications capacity [1]. An important performance parameter for any communications system is the bit error rate (BER), which depends upon the signal-to-noise ratio (SNR) at the receiver’s output. An approach to increasing the SNR of an OWCs or VLCs system is to use silicon photomultipliers (SiPMs) in the receiver [2,3,4,5,6,7,8,9,10,11,12,13,14,15,16,17,18]. These devices are arrays of microcells, containing a single-photon avalanche diode (SPAD), that are designed so that an output pulse is generated whenever an avalanche event occurs. Since a single photon can initiate an avalanche, these microcells can detect individual photons. It is this ability to detect photons that means that the sensitivity of a SiPM receiver is expected to be limited by Poisson statistics.

When on–off keying (OOK) is used the number of detected photons per bit when a zero is received determines the average number of photons per bits required to achieve a particular BER. Since avalanche events can be initiated in the dark, at a rate known as the dark count rate (DCR), existing SiPMs are particularly suited to data rates of more than 100 Mbps. Consequently, SiPM receivers have been shown to operate within a few photons per bit of the noise floor determined by Poisson statistics [5]. Unfortunately, after a microcell has detected a photon, the avalanche event has to be quenched by reducing the voltage across the avalanche photodiode (APD) in the microcell. The microcell then has to be recharged, and this means that the SiPM has a non-linear response [4].

The performance of receivers containing commercially available SiPMs can be determined experimentally [3,4,5,6,7,9,10,11,12,13,14,15,16,17,18]. Ideally, the results of these experiments could be used to inform the selection of the best SiPMs from amongst those that have different characteristics. Unfortunately, the transmitter can have a significant impact on any experimental results. Furthermore, it can be very difficult to separate the impact of SiPMs’ bandwidths and their non-linear responses. Recently, these problems led to the development of a Monte Carlo simulator [19]. The parameters in this simulator were obtained from either a relevant data sheet or experimental results. The results of the simulator were then validated by comparing them with the results of experiments on the impact of ambient light on the performance of receivers [19].

The aim of this paper is to use this Monte Carlo simulator to compare the impacts of SiPMs’ non-linearity and bandwidth on their performance in receivers. The results of this comparison are then used to estimate the performance of commercially available SiPMs when the data are represented by OOK. Guidelines on the selection of SiPMs for use in receivers have been published previously [20,21]; however, both of these papers assumed that the modulation scheme was orthogonal frequency division multiplexing (OFDM), which is not as energy-efficient as OOK [22]. Furthermore, it was assumed that it is possible to increase the areas of SiPMs without changing any other parameters. In contrast, this paper concentrates upon OOK. In addition, the data sheets of SiPMs show that increasing the areas of SiPMs has a significant negative impact on the width of SiPMs’ fast output pulses and hence their bandwidths. Fortunately, there is a method of combining SiPMs so that they act in parallel but retain their bandwidth. The results of the simulation are therefore used to investigate the impact of using multiple SiPMs in parallel in a receiver. Finally, a review of recently developed technology leads to the conclusion that using two stacked integrated circuits to make SiPMs will dramatically increase the data rates that can be supported.

This paper is organized as follows: Section 2 contains a list of the important characteristics of commercially available SiPMs. This is followed by a brief justification of some assumptions made when constructing the Monte Carlo simulator, as well as a description of the small changes that have been made to the simulator, which has previously been reported in detail [19]. Section 3 then starts with a discussion of the BER of an OOK signal when the number of detected photons per bit is determined by Poisson statistics. This section also contains evidence that the equation that represents SiPMs’ non-linearity, previously observed in the presence of ambient light, is also relevant when high irradiances are used to transmit OOK data. Finally, the section contains a discussion of intersymbol interference (ISI) caused by SiPMs’ output pulses and the resulting increase in the rate at which photons need to be detected at different data rates. Section 4 then contains a discussion of the impact of SiPMs’ non-linearity and the width of SiPMs’ fast output pulses. This discussion includes a suggested method for determining the data rate at which each of these phenomena will become important for each SiPM. However, results show that SiPMs with a high maximum count rate can support data rates that are significantly higher than these two data rates. A method that allows SiPMs to be used in parallel and the possible consequences of using this method are then discussed. This is followed by a discussion of the possible advantages of using 850 nm transmitters rather than 405 nm transmitters. The possible performance of a stacked receiver is then discussed. Finally, Section 5 contains some conclusions from this work.

## 2. Materials and Methods

### 2.1. Characteristics of Commercially Available SiPMs

SiPMs are available from AdvanSiD, Broadcomm, First Sensor, Hamamatsu, and onsemi. All of these manufacturers supply SiPMs that contain arrays of microcells, with each microcell containing an APD in a series with a resistor. These microcells are connected in parallel and biased above the breakdown voltage of the APD. This means that, if a photon initiates an avalanche event in a microcell, the resulting current causes a voltage drop across the resistor. The resulting reduction in the voltage across the APD then quenches the avalanche event. The capacitance within the microcell is then recharged until the bias voltage across the APD is restored. During this recovery time the microcells’ ability to detect photons is reduced [19]. Consequently, the need to recharge the microcells creates a non-linear SiPM response [19]. An important difference when selecting SiPMs for receivers is that onsemi creates an output that is capacitively coupled to each microcell. It is this capacitive coupling that creates fast output pulses on this second output. These narrower, fast output pulses are typically an order of magnitude narrower than the output pulses of other SiPMs. Since this significantly reduces ISI at higher data rates, SiPMs from onsemi have often been used to create receivers [5,6,7,10,11,12,13,14,15,16,17,18].

Onsemi produce three series of SiPMs: The C and J series are manufactured on p-on-n substrates [23] and have a peak PDE at approximately 425 nm. The difference between these two series is that the J series uses a through-silicon via (TSV) process to minimize the dead space between microcells. In addition, onsemi produce the RB series of SiPMs. Unlike the other two series, the RB SiPMs are manufactured using an n-on-p substrate [23]. This change in substrate means that the RB devices have a peak PDE at approximately 600 nm. The important parameters for the RB, C, and J series SiPMs sold by onsemi were obtained from the relevant data sheets and are listed in Table 1. In this table, the PDE at 405 nm has been listed because this is the wavelength that has been used to reduce the impact of ambient irradiance from white LEDs [12].

The data for the C and J series devices with the same area and microcell size, for example, the J30035 and C30035 devices, show that the TSV process increases SiPMs’ PDE and number of microcells, as well as significantly reducing their recovery time. This means that when the transmitter operated at 405 nm the J series has been favored [5,6,7,10,11,12,14,15,17,18].

As expected, within any series of SiPMs, reducing the microcell size increases the number of microcells per unit area. However, the data in Table 1 show that this also reduces the PDE. The higher PDE of the larger microcells is an obvious advantage; however, the need to recharge microcells after quenching creates a non-linear response. This means that, at higher photon count rates, a larger number of microcells will be more important than the PDE. The best choice of microcell size will therefore depend upon the expected count rate of photons.

Most importantly, in free space, a transmitter generates particular irradiances at different relative locations [12]. The irradiance required to support a particular data rate and BER is therefore the most important performance metric for free-space receivers. For any irradiance a larger SiPM will detect more photons from the transmitter. Using a larger area should therefore improve a receiver’s performance; however, the data in Table 1 show that increasing the area of a SiPM also increases its fast output pulse width. This means that smaller SiPMs are expected to be able to operate at higher data rates before their output pulses cause ISI. The trade-off in pairs of characteristics, such as the number of microcells per unit area and PDE or SiPM area and fast output pulse width, has been investigated by numerical simulations.

### 2.2. Numerical Simulations

Numerical simulations have been performed by using a Monte Carlo simulator, which has been described in detail previously [19]. This simulator does not include the effects of dark counts, afterpulsing, and optical cross-talk. Previously, it was shown that omitting these phenomena from the simulator did not impact the ability to predict the impact of ambient light on a SiPM [19]. These phenomena have not been added to the current simulation. In the case of the dark counts this is because, for data rates of 1 Gbps or higher, the dark count rates of the simulated SiPMs are negligible compared to the rate at which photons are detected [19]. Similarly, for the SiPMs that are simulated, both the afterpulsing probability and the probability of optical cross-talk are less than 10%. These phenomena might therefore increase the irradiance falling on a SIPM required to achieve a particular performance by up to 10%. However, this is a small increase compared to the impact of the non-linearity and finite pulse width that is the subject of the current study.

The original implementation of the simulator was used to investigate the impact of ambient light. This meant that photons were detected when a zero was received. In contrast, the simulations in this paper assume that the SiPM has been protected from ambient light by using a filter. This means that, when a zero is received, the dominant noise source is the noise from the electronics, for example, the RF amplifier, in the receiver. After amplification, the peak-to-peak voltage for a single avalanche event in a J 30,020 SiPM was 15 mV_pp_. When the beam from the transmitter is blocked a 5 mV_pp_ (three standard deviations) white noise signal was observed. For the simulations whose results are reported in this paper, Gaussian white noise with a peak-to-peak amplitude of one-third of the peak-to-peak amplitude of an avalanche event was added to the output of the simulator before decoding.

## 3. Results

### 3.1. Poisson Statistics and BER

The dominant noise source in a SiPM receiver is expected to be Poisson noise. If this is the case and an OOK signal is transmitted, the BER can be calculated using the following equation [6]:(1)$$BER=\frac{1}{2}\left[\overset{n_{T}}{\underset{k=0}{\sum }}\frac{{\left(N_{Tx}+N_{b}\right)}^{k}}{k!}.e^{-\left(N_{Tx}+N_{b}\right)}+\overset{\infty }{\underset{k=n_{T}}{\sum }}\frac{{\left(N_{b}\right)}^{k}}{k!}.e^{-N_{b}}\right]\;$$
where $N_{Tx}$ is the number of additional detected photons per bit when a one is received, and $N_{b}$ is the average number of photons detected per bit time when a zero is received.

In addition, $n_{T}$ is the threshold used to differentiate a zero from a one. The value of $n_{T}$ that is used is the value that minimizes the BER.

One important consequence of Equation (1) is demonstrated in Figure 1. The results in this figure show that the BER is very sensitive to the number of detecting photons per bit. In addition, this figure shows a linear fit to the relationship between the BER and $N_{Tx}$ obtained using the linear fitting tool in MATLAB. This fit means that, at least in this range of BERs, the relationship between BER, BER, and $\;N_{Tx}$ is as follows:(2)$$N_{Tx}=-\left(log_{10}\left(BER\right)+0.33\right)/0.42$$

The relationship between $N_{b}$ and the average value of $N_{Tx}$, determined using the optimum thresholds, is shown in Figure 2. One important feature of Figure 2 is that $N_{Tx}$ is independent of $N_{b}$ for small values of $N_{b}$. The other important feature is that $N_{Tx}$ is proportional to the square root of $N_{b}$ at high values of $N_{b}$ [18]. Previously, these features have been used to predict that, when ISI is negligible, increasing the area of a SiPM by a factor, f, will reduce the irradiance needed to achieve a particular BER by a factor between f^1/2^ and f [18]. Unfortunately, the results in Table 1 show that increasing the area of a SiPM also increases the width of the output pulses. This strategy may therefore only be successful at lower data rates.

In the past, optical filters have been used to reduce the impact of ambient light; however, these filters will not protect a SiPM from one non-ideal characteristic of some transmitters. In particular, transmitters need a wide bandwidth to support data rates of several Gbps, and wide bandwidths are achieved by not turning off transmitters when they are transmitting a zero. The ratio between transmitters’ output powers when transmitting a zero, $\;P_{0}$, and when transmitting a one, $P_{1}$, is characterized by the extinction ratio (*EXR*):(3)$$EXR=P_{1}/P_{0}\;$$

The results in Figure 3 show that the extinction ratio can have a significant impact on the number of photons per bit needed to achieve a particular BER. Previously, the extinction ratio of a L405P20 405 nm laser diode used to transmit OOK data rates of less than 2.4 Gbps was found to be 15. The impact of this extinction ratio had to be taken into account when predicting the performance of a system accurately [18]. However, at lower extinction ratios, for example, those less than five, the number of photons per bit increases very rapidly as the *EXR* reduces. The *EXR* of a transmitter could therefore play a key role in determining the performance of a system.

### 3.2. Impact of Non-Linearity on BER

The count rate for a SiPM, such as a J30020, can be related to the irradiance of monochromatic light falling on the SiPM, L, via Equation (4):(4)$$C_{rate\;}=N_{cells}\alpha L/\left(1+\alpha \tau_{p}L\right)\;\;$$
where $N_{cells}$ is the number of microcells in the SiPM and $\tau_{p}\;$is a characteristic time. In addition, parameter α is as follows:(5)$$\alpha =\eta \left(V_{ov},\;\lambda \right)A_{\mu }/E_{p}$$
where $\eta \left(V_{ov},\;\lambda \right)\;$is the PDE of the SiPM at a particular over-voltage, $V_{ov}$, and wavelength, $E_{p}$ is the energy of a photon at this wavelength, and $A_{\mu }$ is the active area of a microcell.

Equation (4) was derived assuming that a microcell cannot detect photons for a time, known as the dead time, whilst it recovers after detecting a photon [4]. This equation has previously been used to explain the relationship between the current needed to sustain the SiPM bias voltage and the monochromatic irradiance falling on the SiPM [11]. However, recent simulations have shown that the assumption that a microcell cannot detect a photon during its dead time is not correct [19]. Fortunately, if $\tau_{p}$ is approximately 2.2 times the recovery time listed in the datasheets and Table 1, Equation (4) can be used to explain the current needed to sustain the SiPM bias voltage. In addition, it has been shown that this non-linear response occurs on the fast output used to create a receiver [19].

Previously, SiPMs’ non-linear responses occurred because SiPMs were exposed to ambient light [19]; however, they will also occur at high transmitter irradiances. An important difference between ambient light and light from the transmitter is that the light from the transmitter is modulated. For a J30020 SiPM, at OOK data rates of 1 Gbps or higher $\tau_{p}$ is greater than 33 times the bit time. Under these conditions the varying irradiance from the transmitter is expected to be indistinguishable from a constant ambient irradiance. It is therefore expected that the non-linearity observed at high ambient light irradiances will also be observed at higher OOK data rates. If this is the case, then the irradiance,$\;L_{NL}$, needed to achieve a particular count rate, $C_{rate}$, can be calculated by rearranging Equation (4):(6)$$L_{NL}=C_{rate}/\left(\alpha \left(N_{cells}-\tau_{p}C_{rate}\right)\right)\;\;$$

The usefulness of (6) has been investigated using Monte Carlo simulations. In these simulations the BERs achieved at different data rates when the SiPM is assumed to have a linear response were first calculated. Then, the BER was calculated when Equation (6) was used to compensate for a non-linearity. The results in Figure 4a show the simulated BER for a J30020 SiPM with an output pulse width of zero. These results show that, if the SiPM is assumed to have a linear response, then the target BER is only achieved for data rates less than 1 Gbps. In contrast, using Equation (6) to determine the irradiance needed to achieve the required count rate maintains the BER at less than the target BER up to 100 Gbps. The deviations in the BER from the target value arise because Equation (6) is not a perfect representation of the non-linearity, and the results in Figure 1 show that the BER is very sensitive to the number of detected photons per bit. Taking these factors into consideration, the results in Figure 4a confirm that the same non-linearity occurs when high count rates arise from either ambient light or high data rates.

The irradiances used to obtain the results in Figure 4a are shown in Figure 4b. These results show that the non-linearity has very little effect on irradiances less than the maximum irradiance that would be available from a 405 nm transmitter in a typical office [12].

Furthermore, these results indicate the possible impact of the finite width of the fast output pulses. In particular, in previous experiments with a J30020, a data rate of 3 Gbps was achieved at an irradiance of approximately 100 mWm^−2^ [11]. In contrast, the results in Figure 4b show that, if the width of the fast output pulses is zero, a data rate of 50 Gbps could be achieved at this irradiance. These simulation results show that SiPMs’ non-linearity is not the factor that has previously limited the data rates that have been achieved.

### 3.3. Impact of Pulse Width

The impact of the finite width of fast output pulses has been investigated by determining the BER at various OOK data rates. In these simulations the non-linearity was taken into account by using Equation (6) to determine the transmitter irradiance which will generate 5.2 additional photons per bit when a one is transmitted. The results for a pulse width of 1.4 ns were included in these simulations to represent a J30020. The results for this pulse width in Figure 5a show that the pulse width has an impact on the BER at data rates of less than 1 Gbps. In particular, when the pulse width is 1.4 ns the BER is 10^−2^ at approximately 600 Mbps; however, the results in Figure 4a show that the non-linearity does not cause the same BER until approximately 35 Gbps. This confirms that the pulse width has a significant impact on the performance of a J30020.

The impact of the ISI caused by the width of the output pulses is expected to depend upon the ratio of the pulse width to the bit time. Since the OOK data rate is inversely proportional to the bit time, this is the same as the product of the data rate and the output pulse width. The results in Figure 5b confirm that the impact of ISI depends upon the ratio between pulse width and bit time. Furthermore, the results in Figure 5b show that the BER only starts to increase rapidly when the pulse width equals the bit time. This means that halving the width of the fast output pulses will double the data rate at which ISI creates a significant power penalty.

### 3.4. The Pulse Width Penalty

The penalty from ISI alone has been determined by simulating a SiPM with the same parameters as a J30020, except that the recovery time was set to 1 ps. This recovery time is significantly shorter than the minimum timestep of the simulations. Consequently, all microcells fully charged at each simulated timestep and SiPMs’ responses are therefore linear. For each simulation the data rate was set. Then, the BER after DFE was evaluated for count rates that achieved BERs between 10^−3^ and 10^−2^. Linear interpolation and this data were then used to estimate the count rate that would give a BER of 3.8 × 10^−3^. These results were then expressed as a count rate penalty, which is the count rate needed to achieve a BER of 3.8 ×10^−3^ divided by 5.2 times the data rate, which is the count rate that is expected to achieve this BER. To include the effect of SiPMs’ non-linearity this process was then repeated for a SiPM with the same recovery time as a J30020.

The results in Figure 6a show that, for data rates less than 2 Gbps, the non-linearity has no impact on the count rate penalty; however, the results in Figure 6b show that the non-linearity increases the irradiance required to obtain the same count rate at 2 Gbps. Furthermore, by 2 Gbps the count rate penalty is approximately 16. This means that the required number of photons per bit when a one is transmitted has increased from 6 at 200 Mbps to 82 at 2 Gbps. 

Some results for a SiPM with the same parameters as a J30020 are missing from Figure 6 because it was not possible to achieve a BER of 3.8 ×10^−3^ at higher data rates. The results in Figure 7 show that at 2.5 Gbps the BER starts to increase as the count rate increases. This creates a minimum BER above the required BER. A minimum BER as the irradiance, and hence count rate, increases has been observed experimentally previously [10]. This previously unexpected behavior was found to be due to a new form of ISI caused by SiPMs’ non-linearity. A comparison between the count rates of these simulations and the maximum count rate of the simulated SiPM suggests that this phenomenon has become important once the count rate is 40% of the maximum count rate.

### 3.5. Improving the Agreement between Experimental and Simulated Results 

Using the parameters of a J30020 created an opportunity for comparing simulation results with experimental results. The possible negative impacts of a transmitter’s bandwidth and/or extinction ratio would explain simulation results that were better than the experimental results; however, the results in Figure 8b predict that higher irradiances are required than are observed experimentally, e.g., the experimental results gave a data rate of approximately 3 Gbps at 100 mWm^−2^, whilst the simulation results suggest that this data rate is impossible. This suggests that the 1.4 ns Gaussian pulses used in the simulator are pessimistic. Reducing the fast output pulse width by a factor of 1.5 means that the simulation predicts a data rate of 3 Gbps at an irradiance of 97 mWm^−2^. It appears that reducing the pulse width in the simulator by a factor of 1.5 improves the accuracy of the simulation results.

## 4. Discussion

### 4.1. Previous Published Work

Previously, the performances of receivers containing J30020 and J30035 have been compared [11]. These two SiPMs have very similar output pulse widths. However, its larger number of microcells and faster recovery time mean that the maximum count rate of a J30020 is four times the maximum count rate of a J30035. This significant difference was the motivation for a previous comparison of receivers containing these two SiPMs [11]. This comparison showed that, at an irradiance of 100 mWm^−2^, changing the SiPM increased the data rate from 2.4 Gbps to 3.0 Gbps. This small increase, relative to the increase in the maximum count rate, can now be seen to arise from the rapidly increasing count rate penalty at high data rates. 

### 4.2. Selecting between Available SiPMs at 405 nm

The results in Figure 4a show that non-linearity alone begins to impact the achieved BER at 1 Gbps. This corresponds to an average count rate of 26 Gcps, which is approximately 5% of the maximum count rate. This strict definition of the end of the linear regions arises from the sensitivity of the BER to the number of detected photons per bit, shown in Figure 1; however, the sensitivity of the BER to the detected photons per bit means that the power penalty for higher count rates will be relatively small. This strict definition of the end of the linear regime therefore simply highlights when the transmitter irradiance may need to be increased slightly to achieve the required BER. 

The simulation results suggest two more important criteria that should be considered when selecting an existing SiPM for incorporation into a receiver. These are as follows: (i)The results in Figure 6a show that the effects of the non-linearity and pulse width are independent if the count rate is less than 40% of the maximum count rate. Even when the pulse width penalty is negligible, the non-linearity would almost double the irradiance required to achieve a particular BER. It may therefore be prudent to expect a SiPM to operate with count rates less than 40% of its maximum count rate.(ii)The results in Figure 6a also show that the count rate power penalty is less than two if the bit time is less than the pulse width. To avoid a significant increase in required irradiance, the transmitted data rate should ideally be less than the data rate whose bit time equals the pulse width.

The results that arise from applying these two conditions, shown in Table 2, show that, if they are applied, the suggested data rates for a J30020 are less than the maximum data rate that has been reported [11]. This is because, as with a C10010 and a C30020, the data rate at which condition (i) is reached for a J30020 is much higher than that required for ISI to double the required irradiance, condition (ii). This means that these SiPMs have the capacity to tolerate an ISI power penalty significantly larger than two. 

A J30020 has more microcells, a higher PDE, and a higher maximum count rate than a C30020. The estimated performances of a J30020 and a C10010 are therefore shown in Figure 8. As expected, the results in Figure 8 show that there large maximum count rates that allow these two SIPMs to operate at data rates at which they incur a significant ISI power penalty; however, it is also clear that, despite having faster output pulses, the smaller area of a C10010 means that it is only expected to perform better than a J30020 at data rates higher than 3 Gbps. At these data rates the performance of a J30020 is limited by a combination of ISI power penalty and the saturation of its non-linear response; however, its smaller area means that a C10010 is only a better choice than a J30020 at irradiances that are not eye-safe [12]. 

### 4.3. Using Diodes to Add SiPMs

Operating at OOK data rates whose bit time is significantly shorter than the duration of SiPMs’ output pulses incurs a significant power penalty. This suggests that the primary characteristic to consider when selecting a SiPM to use at higher data rates, e.g., OOK rates about 2 Gbps, should be the duration of the fast output pulses. Unfortunately, the results in Table 1 show that these SiPMs have a smaller area and therefore fewer microcells. The results in Figure 8 show that this means that a J30020 is a better choice than a C10010 at eye-safe irradiance; however, it is possible to use a pair of Schottky diodes on the fast output of each SiPM to add their fast output pulses without increasing the width of the output pulses [27]. Figure 9 shows a schematic diagram of this idea, where Skyworks SMS7621 24 GHz Schottky diode pairs are used to combine the outputs of the SiPMs. In this circuit the SiPMs are biased by connecting a bias voltage, V_bias_, to their cathode. The fast output of each SiPM is connected to the center of a pair of Schottky diodes, which are forward-biased by the bias source, V_ss_, such that each diode pair passes approximately 1 mA. An output pulse on one SiPM will cause the current through the associated diode pair to vary. This current then flows on a common line where it is added to the current flowing from other SiPMs. Any variation in this total current is converted into a voltage by the resistor connected to V_ss_. The high-frequency content of this voltage passes through a capacitor to Fast+, which is the shared output from all of the SiPMs.

The potential advantages of using more SiPMs with narrow fast output pulses has been investigated by assuming that it is possible to increase the number of microcells in a C10010 and a C10020 without changing any of the other parameters. The results in Figure 10 confirm that increasing the effective area of a SiPM reduces the irradiance required to support a particular data rate. The results for the combination of nine C10020 SiPMs show that they might be the best choice for data rates up to approximately 3 Gbps; however, their significantly lower maximum count rate then causes a rapid increase in the required irradiance at approximately the data rates in Table 2. In contrast, their large maximum count rates mean that both a J30020 and the combination of nine C10010 SiPMs can operate a data rates well above those in Table 2. The difference now is that an increase in area means that the combination of C10010 SiPMs is a better choice at data rates of approximately 1.5 Gbps, which is close to the data rate at which ISI caused by the fast output pulses is expected to impact the performance of a J30020.

The possible benefits of using even more SiPMs in parallel are shown in Figure 11. This figure confirms that adding more SiPMs in parallel will reduce the irradiance required to support a particular data rate. The ideal conditions assumed in these estimates, for example, the absence of ambient light and an infinite extinction ratio on the transmitter, mean that the same number of photons per bit have to be detected. This means that, at low data rates, the required irradiance is inversely proportional to the number of SiPMs; however, the important comparison is the data rate that can be supported at a particular irradiance. At irradiances of approximately 4 mWm^−2^ an array of 25 C10010 SiPMs is expected to support 2.8 Gbps, whilst an array of 100 C10010 SiPMs would support 3.5 Gbps. These results show that a rapid rise in the count rate penalty at high data rates limits the performance improvements that can be achieved by increasing the number of SiPMs acting in parallel.

### 4.4. Exploiting the Existing Parallel Fast Outputs

A problem with using multiple SiPMs is that their price is not proportional to their area. Consequently, an array of SiPMs would cost significantly more than a single SiPM with the same area. An alternative way of reducing the width of fast output pulses is suggested by a close inspection of the back side of the larger SiPMs produced by onsemi. This inspection shows that these SiPMs have multiple fast outputs that are connected together to create one fast output [25]. In particular, TSV processes are used at several locations on a SiPM to connect fast outputs for different areas of a SiPM to its bottom side. These fast outputs are then connected together by metal traces. The resulting combined fast output is then connected to a single output pad. Figure 12a shows that a 3 mm by 3 mm J series SiPM has six fast outputs that are connected together to a single pad [28]. Using a connection for each of these areas would create an array of six SiPMs with an area of 1.5 mm^2^ each. These six outputs could be made available separately by a relatively small change at the end of the manufacturing process. They could then be combined by using the method in Figure 9. The result would be a 9 mm^2^ SiPM with a fast output width of less than 1 ns.

### 4.5. Selecting between SiPMs for NIR

Experiments have been performed with 405 nm light to limit the impact of ambient light from white LEDs; however, this choice of wavelength limited the eye-safe power limit [12]. Changing to 850 nm would increase the eye-safe power limit by a factor of approximately 50 and mean that existing high-bandwidth 850 nm transmitters could be used.

The RB series of SiPMs have a higher PDE at 850 nm than the J series SiPMs. This would suggest that they are the better choice for operation with 850 nm transmitters; however, as shown in Table 3, they also have broader fast output pulses. The potential performance of a J30020 and arrays of nine RB10010 SiPMs and nine RB10020 SiPMs is shown in Figure 13. All of these systems have the same overall area. Consequently, their higher PDE means that the two systems made from RB series SiPMs support the lower data rates at lower irradiances than a J30020; however, at data rates of more than 1 Gbps the narrower output pulses of a J30020 mean that the performance of this SiPM is expected to be similar to the performance of the array of RB series SiPMs. Cost would then favor a J30020.

At 405 nm an eye-safe transmitter in a typical office would deliver 3 mWm^−2^ at the edge of its coverage area [12]. Changing to 850 nm would increase this to approximately 150 mWm^−2^. The results in Figure 13 suggest that, at this irradiance, the SiPMs would support approximately 2 Gbps. This is a little higher than the data rate, 1.4 Gbps, achieved using a 405 nm transmitter in the same scenario; however, the results with a 405 nm transmitter were obtained in 500 lux of ambient light using filters to protect a SiPM from ambient light. Despite the increase in eye-safe power, a change to 850 nm is therefore not expected to support significantly higher data rates.

### 4.6. Designing Application-Specific SiPMs

The results show that any new application-specific integrated circuit (ASIC) designed to act as a receiver should have significantly faster output pulses than existing SiPMs. This emphasis suggests that the microcells in the ASIC might have a digital output. SiPMs with digital outputs have been integrated into receivers previously [29,30]; however, both of the SiPMs used in these experiments were created on one chip. This meant that the digital logic circuits alongside each SPAD reduced the overall fill factor, and hence the PDE, of the SiPMs.

Fortunately, since these SiPMs were manufactured the ability to stack two chips has been developed [31]. If one of these chips is used to create an array of APDs and the other to create a matching array of ancillary circuits, then this technology avoids the trade-off between circuit complexity and fill factor. In addition, this technology means that the two chips can be made by using the manufacturing processes best suited to their function. These advantages mean that stacked systems, often known as SPAD arrays, have been created in which one of these chips contains an array of APDs and the other chip contains a matching array of quenching circuits as well as relatively sophisticated digital circuits [31].

The existing stacked SPAD arrays have been designed for applications such as LIDAR and low-light imaging; however, they contain components that could be used to create a receiver. The most important component which could be used in a receiver is the APDs in the first chip. These have been made by using a variety of manufacturing processes, and some include additional features that increase the PDE at some or all wavelengths. One feature used in some devices is a charge-focusing SPAD, in which the electric field in the APD guides photon-generated electrons into a central avalanche region [32]. This approach results in a fill factor of 100% and a PDE of approximately 40% at 405 nm. Furthermore, SPADs can be created with a 6.39 µm pitch and in arrays of 2072 by 1548 [32].

A key function of the ancillary circuits in the second chip is to quench the otherwise self-sustaining avalanche processes. In order to reduce afterpulsing, this can be performed by combining a passive quenching process with an active reset (a combination known as PQAR). To achieve this, a MOSFET is connected in series with the APD to create a load that reduces the bias voltage across the APD when an avalanche occurs. The resulting change in the voltage across the APD is then detected by a digital circuit. This circuit is designed to hold the APD bias voltage below the breakdown voltage for a controlled time. During this time, any charge trapped in the APD can escape without creating an afterpulse; however, the cost of suppressing afterpulses is that the SPAD cannot detect a photon, and so this time is the dead time for the SPAD. At the end of the dead time the digital circuits rapidly reset the APD bias voltage. Since this minimizes the probability that a photon is detected whilst a SPAD is being recharged, it reduces the risk that the SPAD can be paralyzed at high irradiances [33]. Depending upon the application, the digital signal generated by an avalanche event can be processed in one of several different ways. An example of the circuit complexity that can be achieved is a low-light image sensor [34]. In this case, the ancillary circuit associated with each SPAD included a quenching circuit, as well as a nine-bit counter to count the detected photons; however, this might overflow, and so it also contained an additional five-bit latch which, together with the nine-bit latch from the counter, can store a fourteen-bit code that represents the time at which the counter overflows. It also includes a 15-bit multiplexer to connect the contents of these 14 latches and an overflow flag to a shared 15-bit bus. All of this functionality was achieved in a 12.24 μm pitch.

This example system indicated the functionality that can be achieved in an area that is smaller than the area of the microcells in most existing SiPMs; however, the challenge when supporting multi-Gbps OOK data rates is that data have to be obtained from each array element in a small fraction of a nanosecond. Fortunately, the location at which a photon is detected is not important. This means that each SPAD can be allowed to transmit its response to a detected photon as soon as it occurs. Furthermore, the method of combining SPAD outputs should support the simultaneous detection of a few photons and different modulation schemes. 

One way to accommodate the simultaneous detection of photons would be to generate a narrow output current pulse when an avalanche occurs. In the exemplar system, the quenching and digital logic circuits were manufactured in a 40 nm CMOS process [34]. A 1 μm-wide transistor manufactured in a 40 nm process can pass a current of approximately 400 μA [35]. In addition, the maximum frequency at which these transistors can amplify a signal is at least 200 GHz [35]. This suggests that it should be possible to generate 10 ps current pulses when a photon is detected. Furthermore, it should be possible to create transimpedance amplifiers on the same chip that create an analogue output voltage that is proportional to the sum of these short current pulses.

Table 4 contains a comparison between a J30020 and a possible stacked SPAD receiver. The two systems are assumed to have the same area. The pitch of the stacked receiver is assumed to be the same as the pitch of the exemplar system [34]. The result is a four-fold increase in the number of microcells. In addition, it is assumed that the dead time of the stacked system is 8 ns. This is the time previously used to suppress afterpulsing [36]. Finally, although it may be possible to create 10 ps output pulses, a conservative estimate of 100 ps is included in the table.

If the assumed fast output pulse width can be achieved, then the stacked system would not suffer from ISI caused by the receiver for data rates of less than 10 Gbps. At lower data rates it would therefore require an average of approximately 2.6 photons per bit to achieve a BER of 3.8 × 10^−3^. A data rate of 10 Gbps would then be achieved when the average irradiance from a 405 nm transmitter is 1.8 mWm^−2^. This is less than the irradiance that can be obtained from an eye-safe 405 nm transmitter in a typical office [12]. This receiver therefore has the potential to deliver data rates of up to 10 Gbps in a typical office.

A potential hurdle to achieving 10 Gbps using 405 nm is the bandwidth of 405 nm transmitters. In this case, it may be prudent to change to 850 nm transmitters designed to deliver data rates of more than 10 Gbps. At 850 nm the SPAD could have a PDE of 30% or more [32]. In this case, the lower energy of 850 nm photons means that 10 Gbps could be achieved at an irradiance of 1.1 mWm^−2^. This would be eye-safe and could be delivered using a 850 nm transmitter that is designed to deliver 10 Gbps or higher [37]. Two or more of these devices could be used separately or in parallel to cover the 2 m by 2 m area covered by a single transmitter in a typical office environment [12]. The result would be an irradiance that could support 10 Gbps.

In the absence of an ISI power penalty from either the transmitter or the receiver, 10 Gbps requires an average count rate of approximately 26 Gcps, which is a tiny fraction of the potential maximum count rate of 7509 Gcps. This maximum count rate suggests that the stacked receiver would have the capacity to tolerate a count rate penalty of 144; however, to avoid the negative impact from the non-linearity it may be more efficient to limit the penalty to half of this value. This would suggest that the receiver could support data rates of 30 Gbps. This is slightly higher than the data rate that is supported by receivers designed for fiber optical communications [37]; however, the existing receivers need 0.3 mW of received optical power to support a data rate of 25.78 Gbps and a BER of 5 × 10^−5^. Under ideal conditions this BER requires an average of approximately five photons per bit; however, the extinction ratio of the transmitter in this system is two, and Figure 3 shows that this means that the average number of photons per bit must be approximately 60. Taking these factors into account, without any ISI penalty the stacked receiver would require only 0.6 μW at the same data rate, wavelength, BER, and extinction ratio. Even with a significant ISI penalty the stacked receiver would therefore be expected to require only a small fraction of the optical power of the existing receiver to support 25.78 Gbps.

Furthermore, the stacked receiver’s dead time in Table 4, 8 ns, is the dead time used to reduce the afterpulsing probability for applications where afterpulses may be a significant problem. In contrast, previous Monte Carlo simulations have replicated experimental results despite the fact that afterpulsing is not included in these simulations. This suggests that it may be possible to reduce the dead time of these systems significantly. The result could be a receiver that can support data rates approaching 1 Tbps.

## 5. Conclusions

Results have been reported which confirm that SiPMs with narrow output pulses should be preferred when selecting SiPMs for incorporation into VLCs or OWCs receivers. Furthermore, the irradiance required to achieve a particular BER increases rapidly once the bit time is shorter than the output pulse width.

Although the pulse width is a very important parameter, SiPMs’ non-linearity must be taken into account. In particular, it is suggested that a SiPM should operate with a count rate of less than 40% of its maximum count rate.

The need to detect a constant number of photons per bit means that increasing the area of a SiPM should reduce the irradiance needed to support a particular data rate. Unfortunately, increasing the area of a single SiPM of a particular type increases its pulse width. The trade-off between area and pulse width can be avoided by using diodes to add the outputs of SiPMs acting in parallel; however, using this method to significantly increase the data rate would be expensive. This cost increase could be reduced by using a small change in the last stages of the manufacturing process of individual larger SiPMs to create single SIPMs with multiple outputs that have narrower output pulses.

Reasons for a change in transmitter wavelength from 405 nm to 850 nm have been highlighted; however, results have been presented which suggest that the benefits of this change will be relatively small.

Finally, a brief survey of systems made by stacking arrays of SPADs onto a second chip has been presented. This survey suggests that this new technology could dramatically improve the performance of receivers. The result would be a receiver that is significantly better than existing receivers for fiber-optic communications operating at 25.78 Gbps. Factors that would make it possible to create receivers operating at significantly higher data rates have been highlighted.

## Figures and Tables

**Figure 1 sensors-23-01101-f001:**
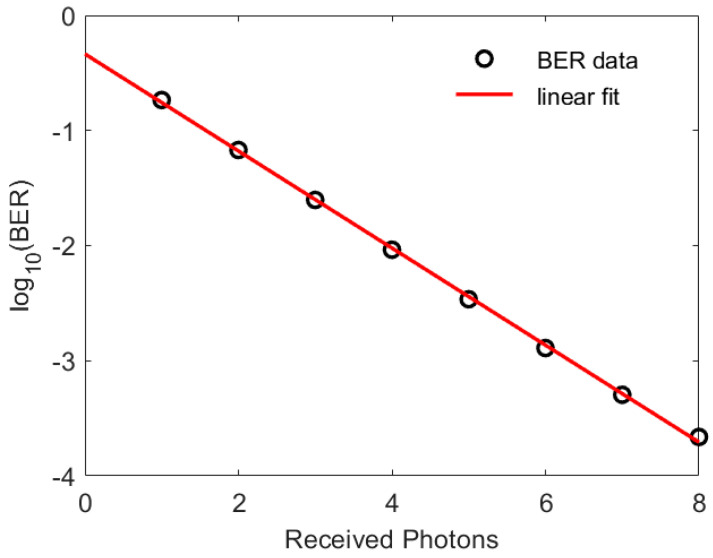
The number of recieved photons required to represent a logic one, $N_{Tx}$, to achieve a range of BERs when $N_{b}=0$. The range of BERs shown is close to the value of 3.8 × 10^−3^ needed when forward error correction is used to significantly reduce the final BER. The results obtained with different integer numbers of additional photons are shown as black circles. The red line is then a linear fit to the points.

**Figure 2 sensors-23-01101-f002:**
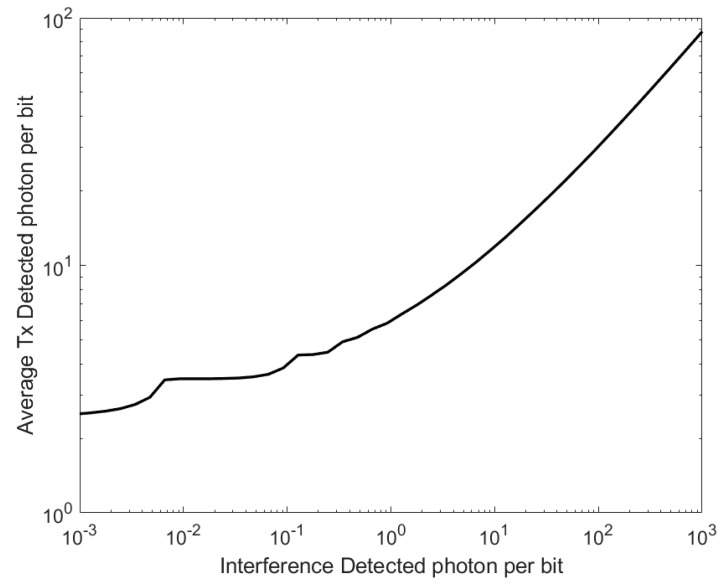
The average number of additional detected photons per bit required to achieve a BER of 3.8 × 10^−3^ when varying numbers of photons per bit are detected when a zero is being received.

**Figure 3 sensors-23-01101-f003:**
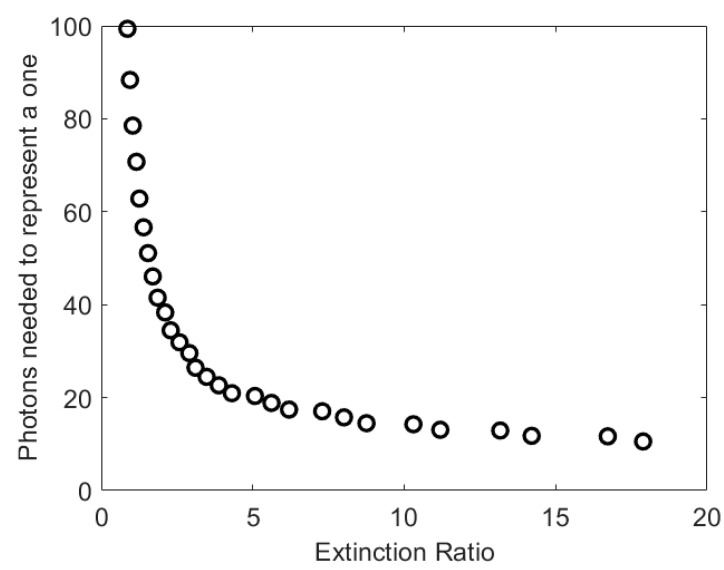
The impact of the extinction ratio of the transmitter on the number of photons per bit needed to represent a one and achieve a BER of 3.8 × 10^−3^.

**Figure 4 sensors-23-01101-f004:**
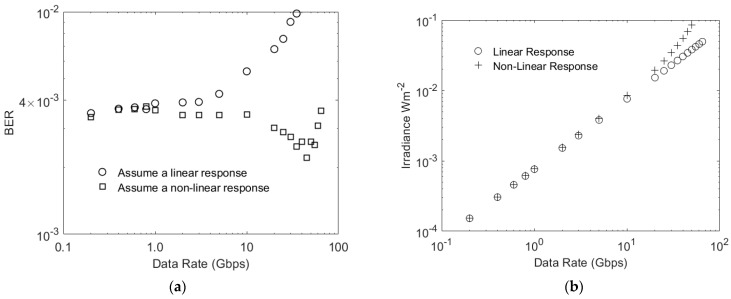
(**a**) The BER from a simulation that assumed a fast output pulse width of zero, and when a zero is transmitted there are no detected photons. This means that, to achieve the target BER of 3.8 × 10^−3^, approximately 5.2 photons have to be detected. (**b**) The irradiances used to generate the results in (**a**).

**Figure 5 sensors-23-01101-f005:**
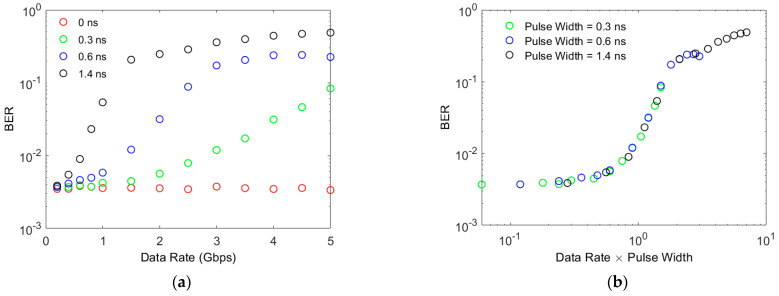
The BER achieved when 5.2 photons are detected per bit when a one is transmitted. (**a**) shows the results at different data rates compared to the ideal result. (**b**) shows that the important parameter is the product of the pulse width and the data rate.

**Figure 6 sensors-23-01101-f006:**
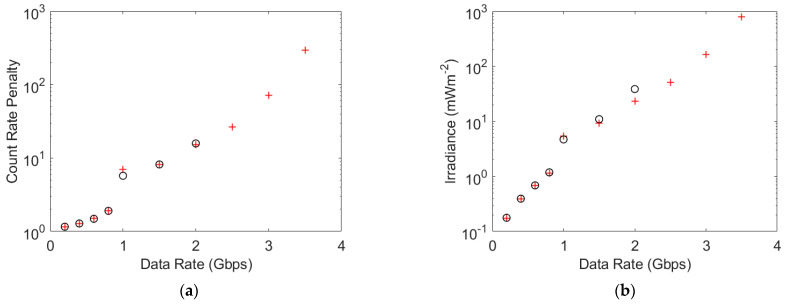
(**a**) The count rate penalty required to achieve a BER of 3.8 × 10^−3^ at data rates that are high enough for the fast output pulses to create ISI. The parameters for a J30020 were used to obtain the simulation results shown in black circles (o). The simulation results shown as red crosses (+) were obtained assuming that a J30020 SiPM had a recharge time of 1 ps, and therefore had a linear response. The irradiance needed by a J30020 to support these count rate penalties is shown in (**b**).

**Figure 7 sensors-23-01101-f007:**
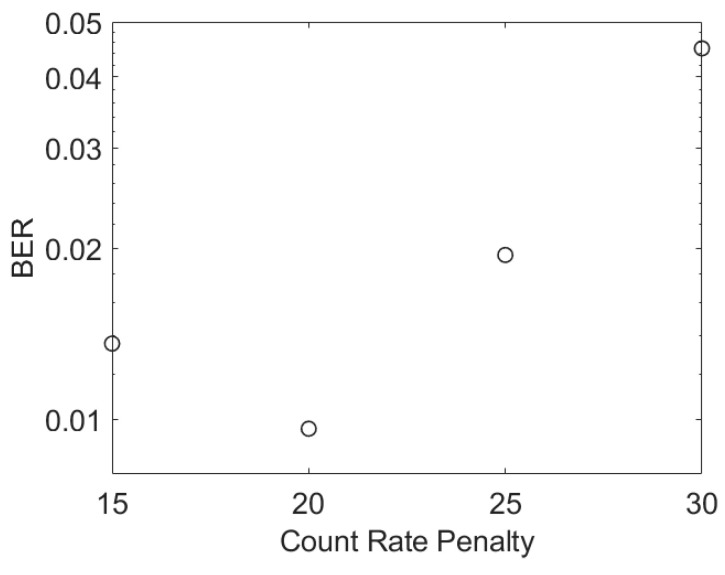
The simulated BER of a SiPM with the same parameters as a J30020 at 2.5 Gbps.

**Figure 8 sensors-23-01101-f008:**
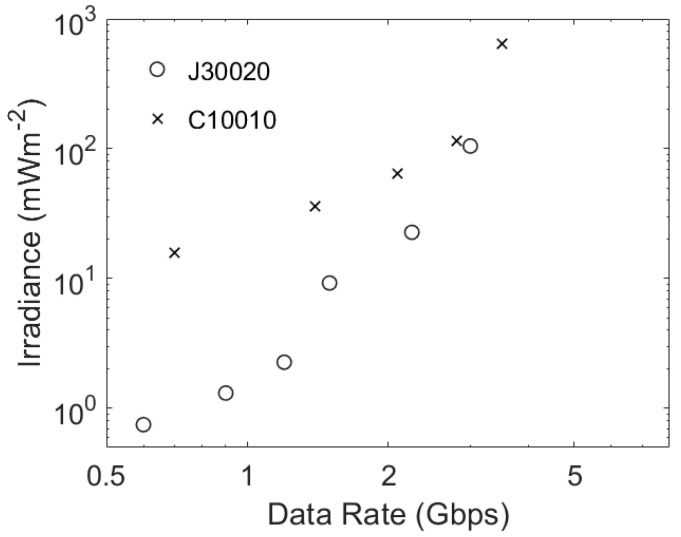
The estimated performance of a J30020, a C1000, and a C10020. These results were obtained by first using the ratio of the bit time to the estimated equivalent Gaussian pulse width to determine the ISI count rate penalty using the results in Figure 7a. This was then converted into the required irradiance using the relevant SiPM parameters and Equation (6).

**Figure 9 sensors-23-01101-f009:**
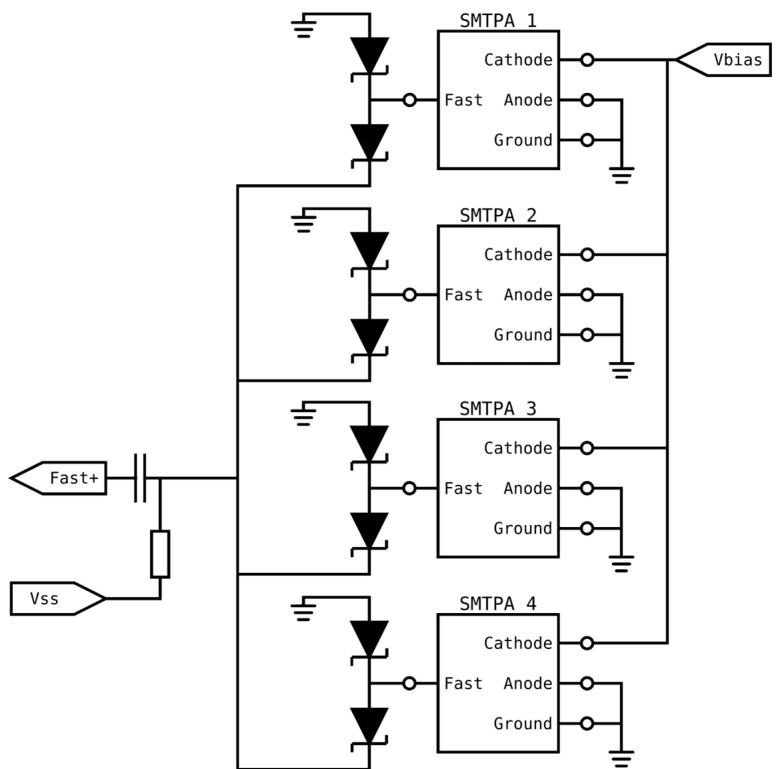
Schematic showing a method of combining multiple SiPM fast outputs together using diode pairs [27].

**Figure 10 sensors-23-01101-f010:**
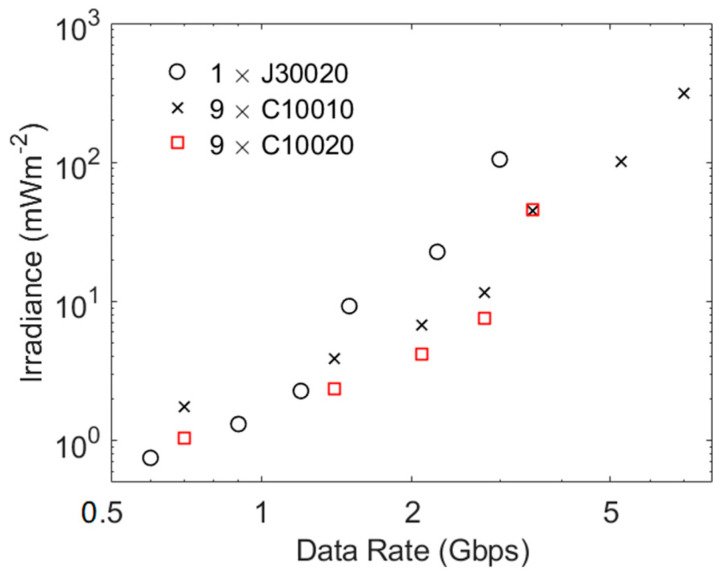
The potential performance of a J30020 and combinations of 9 C10010 SiPMs as well as 9 C10020 SiPMs. These results were obtained using the same methodology as used to obtained the results in Figure 9.

**Figure 11 sensors-23-01101-f011:**
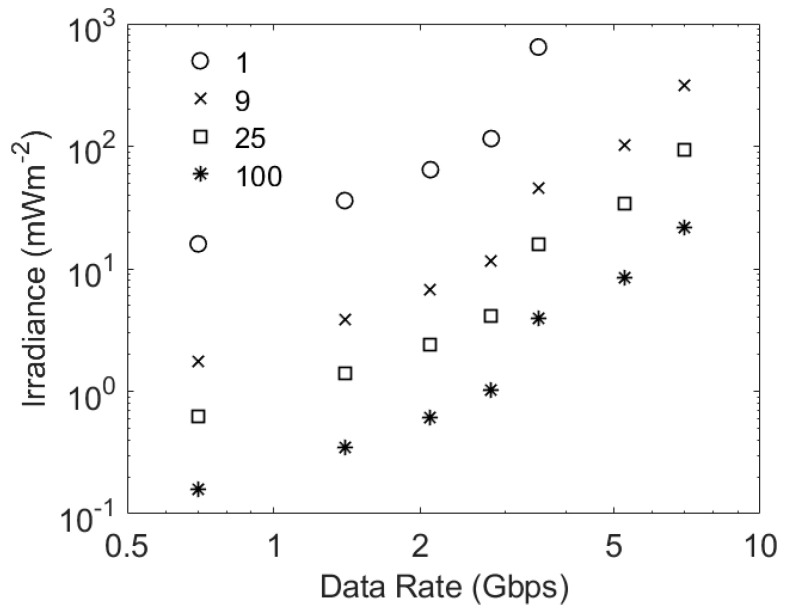
The potential performance of varying numbers of C10010 SiPMs working in parallel. These results were obtained using the same methodology as that used to obtained the results in Figure 9.

**Figure 12 sensors-23-01101-f012:**
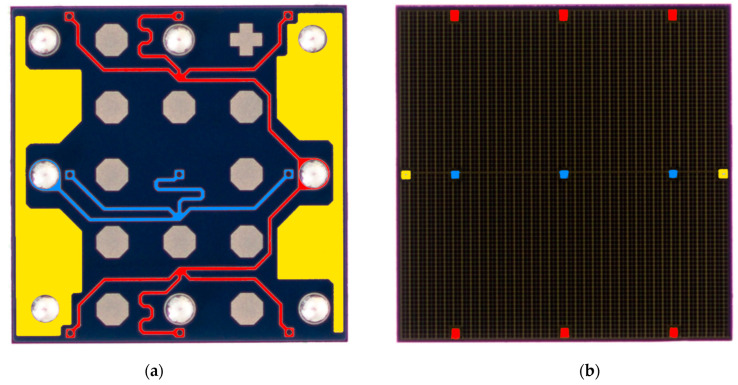
A view of a J series 30035 SiPM. (**a**) shows the back of the SiPM, where different traces are highlighted different colors for visibility. Yellow is the cathode (connected to bias source), blue is the anode (connected to ground, or a series resistor to measure instantanous bias current). Red traces are the fast outputs. The fast output on this device is combined from six separate regions. (**b**) shows the top of the SiPM, where the through-silicon via processes connect to the traces on the rear. (Adapted from [28]).

**Figure 13 sensors-23-01101-f013:**
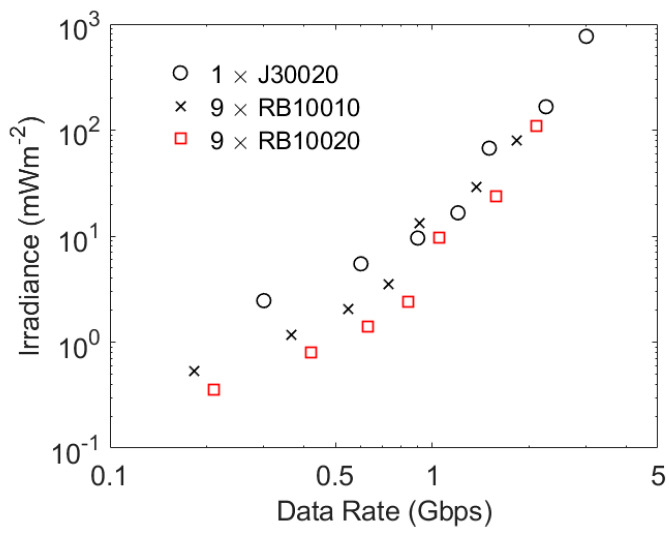
The potential performance of a J30020 and combinations of 9 RB10010 SiPMs and 9 RB10020 SiPMs. These results were obtained using the same methodology as used to obtained the results in Figure 8.

**Table 1 sensors-23-01101-t001:** Key parameters for the three series of commercially available SiPMs manufactured by onsemi [24,25,26].

Name	Area (mm^2^)	Pitch (μm)	Number of μcells	Recovery Time (ns)	Maximum Count Rate (Gcps)	PDE at 405 nm	Fast Output Pulse Width (ns)
RB10010	1	10	4296	12	162.7	0.1	2.3
RB10020	1	20	1590	21	34.4	0.11	2
RB10035	1	35	620	73	3.9	0.12	3.7
C10010	1	10	2880	5	261.8	0.17	0.6
C10020	1	20	1296	23	25.6	0.29	0.6
C10035	1	35	504	82	2.8	0.39	0.6
C30020	9	20	10,998	23	217.4	0.29	1.5
C30035	9	35	4774	82	26.5	0.39	1.5
C30050	9	50	2668	159	7.6	0.44	1.5
C60035	36	35	18,980	95	90.8	0.39	3.2
J30020	9	20	14,410	15	436.7	0.38	1.4
J30035	9	35	5676	45	57.3	0.46	1.5
J40035	16	35	9260	48	87.7	0.46	1.7
J60035	36	35	22,292	50	202.7	0.46	3

**Table 2 sensors-23-01101-t002:** Two important parameters of the onsemi SiPMs together with the data rates determined by the two criteria described in the text.

Name	Maximum Count Rate (Gcps)	Fast Output Pulse Width from the Data Sheet (ns)	Estimated Equivalent Gaussian Fast Output Pulse Width (ns)	Data Rate (i) (Gbps)	Data Rate (ii) (Gbps)
RB10010	162.7	2.3	1.5	12.5	0.7
RB 10020	34.4	2.0	1.3	2.6	0.8
RB 10035	3.9	3.7	2.5	0.3	0.4
C10010	261.8	0.6	0.4	20.1	2.5
C10020	25.6	0.6	0.4	2.0	2.5
C10035	2.8	0.6	0.4	0.2	2.5
C30020	217.4	1.5	1.0	16.7	1.0
C30035	26.5	1.5	1.0	2.0	1.0
C30050	7.6	1.5	1.0	0.6	1.0
C60035	90.8	3.2	2.1	7.0	0.5
J30020	436.7	1.4	0.9	33.6	1.1
J30035	57.3	1.5	1.0	4.4	1.0
J40035	87.7	1.7	1.1	6.7	0.9
J60035	202.7	3.0	2.0	15.6	0.5

**Table 3 sensors-23-01101-t003:** Key parameters for the two series of commercially available SiPMs manufactured by onsemi, including their PDE at 850 nm [22,24,25].

Name	Area (mm^2^)	Pitch (μm)	Number of μcells	Recovery Time (ns)	Maximum Count Rate (Gcps)	PDE at 850 nm	Fast Output Pulse Width (ns)
RB10010	1	10	4296	12	162.7	0.07	2.3
RB10020	1	20	1590	21	34.4	0.12	2
RB10035	1	35	620	73	3.9	0.17	3.7
J30020	9	20	14,410	15	436.7	0.025	1.4
J30035	9	35	5676	45	57.3	0.03	1.5
J40035	16	35	9260	48	87.7	0.03	1.7
J60035	36	35	22,292	50	202.7	0.03	3

**Table 4 sensors-23-01101-t004:** Comparison between a J30020 and a potential stacked SPAD receiver.

Name	J30020	Stacked
Area (mm^2^)	9	9
Pitch (μm)	20	12.24
Number of μcells	14,410	60,073
Recovery time/dead time (ns)	15	8
Maximum count rate (Gcps)	436.7	7509
PDE at 405 nm	0.38	0.4
Fill factor	0.62	1
Fast output pulse width (ns)	1.4	0.1

## Data Availability

The data presented in this study, and the accepted version of this paper, are openly available in the Oxford Research Archive (ORA), https://doi.org/10.5287/bodleian:ORodKRbm0 (accessed on 13 January 2023).

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
