# Peer review of "A Roadmap for Gigabit to Terabit Optical Wireless Communications Receivers"

_sensors, 2023, doi:10.3390/s23031101_

Round 1

Reviewer 1 Report

The work is really interesting. I am sure it will have lots of applications. Good luck for future works.

Author Response

The work is really interesting. I am sure it will have lots of applications. Good luck for future works

We thank the reviewer for their comment on the work and best wishes.

Reviewer 2 Report

Thank you for the nice and important paper. I think the paper presents some important results and conclusions towards improving the bandwidth in optical wireless communication receivers which is worth publication for interested authors. I did have a few questions:

1) the effect of the fast output pulse width  is studied and modeled in their simulation to study the BER taking into account the non-linearity in the model introduced by equation 6. What is unclear to me in the presentation if this is simply a parameterization that fits the observed rates or if there is an underlying theory of why it has this form or simply based on empirical observation. Perhaps there is something simple that I am just missing, and if so I apologize. However, it would greatly improve the clarity if this could be made explicit.

2) at the end of paper the authors suggest "These six outputs could then be combined using the method in Fig 10. Can that actually be done to verify it works experimentally?

3) please check the consistency of capitalization and abbreviations . I did not see ISI defined in line 99 and onsemi is used both as onsemi and Onsemi (i think the logo is lower case but as a proper name is caps - should chose one but both are used in the table 1 caption and the text)

Author Response

Reviewer 2

Thank you for the nice and important paper. I think the paper presents some important results and conclusions towards improving the bandwidth in optical wireless communication receivers which is worth publication for interested authors.

We thank the reviewer for these comments on the importance of the paper.

I did have a few questions:

1) The effect of the fast output pulse width  is studied and modeled in their simulation to study the BER taking into account the non-linearity in the model introduced by equation 6. What is unclear to me in the presentation if this is simply a parameterization that fits the observed rates or if there is an underlying theory of why it has this form or simply based on empirical observation. Perhaps there is something simple that I am just missing, and if so I apologize. However, it would greatly improve the clarity if this could be made explicit.

Equation (6) is derived from equation (4). This wasn’t made clear in the existing text and so the following text has been added just before (6) ‘can be calculated by rearranging (4) to give’

2) at the end of paper the authors suggest "These six outputs could then be combined using the method in Fig 10. Can that actually be done to verify it works experimentally?

This experimental validation would require a relatively small change in the manufacturing process. To make this clear the following sentence has been added just before the sentence highlighted by the reviewer. In particular, the sentence ‘These six outputs could be made available separately by a relatively small change at the end of the manufacturing process

3) please check the consistency of capitalization and abbreviations . I did not see ISI defined in line 99 and onsemi is used both as onsemi and Onsemi (i think the logo is lower case but as a proper name is caps - should chose one but both are used in the table 1 caption and the text)

Unfortunately, we hadn’t checked the spelling of the company name in the captions for two tables. These have now been changed. The only place in which a capital letter is used is at the start of a sentence.  

Reviewer 3 Report

All comments are reported in the attached file.

Please, pay attention and consider all the main comments and suggestions.

Author Response

Figure 4 and Figure 5 are too far from the text and eventually they could be grouped together, considering the function of Fig. 5

               These figures have been moved.

Figures 6 (a) and (b) can be aligned as left-right instead of topbottom. The same for figures 7 (a) and (b).

               The figure are now side by side.

The removal of Figure 5 required the later figures to be renumbered and all references to these figures to be up-dated.

Please, explain somewhere how Dark Count Rate (DCR), Direct CrossTalk (DiCT), Delayed Cross Talk(DeCT), AfterPulses (AP) could affect the response in terms of non-linearity and/or penalty. And/or why, these parameters have not been considered.

The omission of these phenomena is explained in [19]. However, this is a very recent paper and so to explain the omission of these phenomena the following text has been added to the paper, ‘This simulator doesn’t include the effects of dark counts, after-pulsing and optical cross-talk. Previously, it was shown that omitted these phenomena from the simulator didn’t impact the ability to predict the impact of ambient light on a SiPM [19]. These phenomena have not been added to the current simulation. In the case of the dark counts this is because, for data rates of 1 Gbps or higher, the dark count rates of the simulated SiPMs are negligible compared to the rate at which photons are detected [19]. Similarly, for the SiPMs that are simulated both the after-pulsing probability and the probability of optical cross-talk are less than 10%. These phenomena might there-fore increase the irradiance falling on a SIPM required to achieve a particular performance by up to 10%. However, this is a small increase compared to the impact of the non-linearity and finite pulse width that are the subject of the current study.’

This aspect could result particularly important in the case of ganged SiPMs for increasing the SiPM area.

‘Ganging’ will increase the dark count rate. However, it will also increase the maximum count rate by the same amount. It will therefore have no impact on the non-linearity. In addition, the increase in dark counts is from such a low starting point that this it will only have a small effect on the irradiance required by the receiver. Furthermore, [19] explains why increasing the area of the SiPM will reduce the irradiance required to support a particular data rate in the presence of dark counts or ambient light.  

After-pulsing and cross-talk are fixed percentages of the number of detected photons. Since the number of photons that must be detected is independent of the area of the SiPM the impact of these phenomena won’t be changed by ganging SiPMs. The impact of these phenomena on ‘ganged’ SiPMs therefore isn’t specifically discussed in the revised paper

A Conclusion paragraph could be appreciated and highlight the results of the present work. Please, use the same style for figures, captions, sections/ subsections titles, etc

We welcome this suggestion because it leads to a significant improvement in the paper. A conclusion has therefore been added. In particular the paper now includes section 5

‘5. Conclusions

Results that have been reported which confirm that SiPMs with narrow output pulses should be preferred when selecting SiPMs for incorporation into VLC or OWC receivers. Furthermore, the irradiance required to achieve a particular BER increases rapidly once the bit time is shorter than the output pulse width. 

Although the pulse width is a very important parameter the SiPM non-linearity must be taken into account. In particular, it is suggested that a SiPM should operate with a count rate less than 40% of its maximum count rate.

The need to detect a constant number of photons per bit means that increasing the area of a SiPM should reduce the irradiance needed to support a particular data rate. Unfortunately, increasing the area of a single SiPM of a particular type increases its pulse width.  The trade-off between area and pulse width can be avoided by using diodes to add the outputs of SiPMs acting in parallel. However, using this method to significantly increase the data rate would be expensive. This cost increase could be reduced by using a small change in the last stages of the manufacturing process of individual larger SiPMs to create single SIPMs with multiple outputs that have narrower output pulses.

Reasons for a change of transmitter wavelength from 405 nm to 850 nm have been high-lighted. However, results have been presented which suggest that the benefits of this change will be relatively small.

Finally, a brief survey of systems made by stacking arrays of SPADs onto a second chip has been presented. This survey suggests that this new technology could dramatically im-prove the performance of receivers. The result would be a receiver that is significantly better than existing receivers for fibre-optic communications operating at 25.78 Gbps. Factors that would make it possible to create receivers operating at significantly higher data rates have been highlighted. ‘

  1. Materials and Methods

2.1 Characteristics of Commercially available SiPMs

-line 88: SensL, AdvanSid and other companies could be mentioned

In many cases SiPMs have been sold by companies that have then been acquired by other companies. We haven’t included SensL because this company was acquired by On Semiconductor, now onsemi, in 2018. (https://www.onsemi.com/company/news-media/press-announcements/en/on-semiconductor-acquires-sensl-technologies-ltd-leading-provider-of-sipm-spad-and-lidar-sensing-products) This means that the products they developed are now sold under the onsemi brand.

The fate of AdvanSid isn’t quite so clear. Its website news hasn’t been up-dated since 2019, but it was acquired by the Cefla group in 2022. Despite this uncertainty we have included this company in the list.

The only other company that emerged from a search for ‘SiPM’ on the internet and a search of the websites of major component suppliers was First Sensor. This company is now part of TE connectivity and their SiPMs are listed on the website of the component supplier Mouser.

The paper has been revised to say ‘SiPMs are available from AdvanSiD, Broadcomm, First Sensor, Hamamatsu and on-semi.’

-line 99-100: These narrower, fast output pulses reduce ISI and therefore SiPMs from onsemi have often been used to create receivers

CCCC>>> Please, try to make a quantitative comparison with respect to the other SiPMs products in terms of fast output pulses

The paper now says ‘. These narrower, fast output pulses are typically an order of magnitude narrower than the output pulses on other SiPMs.’

  1. Results

3.2 Impact of non-linearity on BER

-line 218-219: The recovery time of the J30020 in Table 1 means that the characteristic time of its non-linearity is 33 ns.

CCCC>>> In Table 1 the recovery time of J30020 is reported to be 15 ns.

Please explain the difference among linear and not-linear.

The difference between these two times isn’t a difference between linear and non-linear. Rather, τp is a characteristic time introduced in equation(4), which has been found to be equal to 2.2 times the recovery time from the data sheet.

The original sentence previously on line 218 has been changed to

‘Fortunately, if  is approximately 2.2 times the recovery time listed in the datasheets and Table 1 [19], (4) can be used to explain the current needed to sustain the SiPM bias voltage.

The text has also been amended to say ‘For a J30020 SiPM, at OOK data rates of 1 Gbps or higher  is greater than 33 times the bit time’ 

3.3 Impact of pulse width

line 258-259: the results in Fig 4 show that the non-linearity doesn’t cause the same BER until approximatley 35 Gbps.

CCCC>>> In Figure 4 (a) the maximum value showed on the X axis is 5 Gbps. It is not clear how it can be stated on 35 Gbps.

A fuller extract from the paper is ‘The results for a pulse width of 1.4 ns were included in these simulations to represent a J30020. The results for this pulse width in Fig. 6 (a) show that the pulse width has an impact of the BER at data rates of less than 1 Gbps. In particular, when the pulse width is 1.4 ns the BER is 10-2 at approximately 600 Mbps. However, the results in Fig 4  show that the non-linearity doesn’t cause the same BER until approximately 35 Gbps.’

There isn’t a Figure 4(a) in the paper. However, the above extract suggests that the reviewer may be looking at Figure 6(a) rather than Figure 4. As the reviewer states the x-axis in Figure 6(a) only extends to 5 Gbps, but the reviewers comments is about  Figure 4 and the x-axis of this figure extends up to 100 Gbps. 

The paper hasn’t therefore been changed in response to this comment. However, in response to other comments Figure 5 has becomen Figure 4 (b) and the entire text has been revised to make it easier to read.

  1. Discussion –

4.2 Selecting between available SiPMs at 405 nm

line 340 - 342: (ii) The results in Fig. 7 (a) also show that the count rate power penalty is 2 at a data the data rate at which the bit time is approximately equal to the pulse width used in these simulations.

CCCC>>> Please, try to ease and explain better the above sentence

In response to another comment the relevant figure is now 6(a)

The text has been amended to say ‘The results in Fig. 6 (a) also show that the count rate power penalty is less than 2 if the bit time is less than the pulse width. To avoid a significant increase in required irradiance the transmitted data rate should ideally be less than the data rate whose bit time equals the pulse width.’

We think that this is much better and thank the reviewer for highlighting the need for improvement.

---> 4.4 Selecting between SiPMs for NIR –

line 461-463: The potential performance of a J30020 and arrays of 9 RB10010 SiPMs and 9 RB10020 SiPMs are shown in Fig. 14 All these systems have the same overall area.

RRRR>>> The potential performance of a J30020 and arrays of 9 RB10010 SiPMs and 9 RB10020 SiPMs are shown in Fig. 14. All these systems have the same overall area. --->

               The missing full stop has been added.

4.5 Designing application specific SiPMs –

line 501: SPADs can be created with a 6.39μm pitch and in arrays of 2,072 by 1,548 [32].

RRRR>>> SPADs can be created with a 6.39 μm pitch and in arrays of 2,072 by 1,548 [32].

               The missing space has been added.

 Please, check carefully space among words and units, typos, etc. A final check is needed before the submission

A final check has been performed and the paper has been revised with the aim of making it easier to read